# Controlled Formation of a Protein Corona Composed of Denatured BSA on Upconversion Nanoparticles Improves Their Colloidal Stability

**DOI:** 10.3390/ma14071657

**Published:** 2021-03-28

**Authors:** Samah Shanwar, Liuen Liang, Andrey V. Nechaev, Daria K. Bausheva, Irina V. Balalaeva, Vladimir A. Vodeneev, Indrajit Roy, Andrei V. Zvyagin, Evgenii L. Guryev

**Affiliations:** 1Institute of Biology and Biomedicine, Lobachevsky State University of Nizhny Novgorod, 603950 Nizhny Novgorod, Russia; samahshanwar@gmail.com (S.S.); bausheva16@mail.ru (D.K.B.); irin-b@mail.ru (I.V.B.); v.vodeneev@mail.ru (V.A.V.); andrei.zvyagin@mq.edu.au (A.V.Z.); 2ARC Centre of Excellence “Nanoscale BioPhotonics”, Department of Physics and Astronomy, Macquarie University, Sydney 2109, Australia; liuen.liang@mq.edu.au; 3Department of Chemistry and Technology of Biologically Active Compounds, Medical and Organic Chemistry, M.V. Lomonosov Institute of Fine Chemical Technologies, MIREA-Russian Technological University, 119571 Moscow, Russia; chemorg@mail.ru; 4Department of Chemistry, University of Delhi, Delhi 110007, India; iroy@chemistry.du.ac.in; 5The Institute of Molecular Medicine, I.M. Sechenov First Moscow State Medical University, 119991 Moscow, Russia

**Keywords:** upconversion nanoparticles, protein corona, colloidal stability, lyophilization

## Abstract

In the natural fluidic environment of a biological system, nanoparticles swiftly adsorb plasma proteins on their surface forming a “protein corona”, which profoundly and often adversely affects their residence in the systemic circulation in vivo and their interaction with cells in vitro. It has been recognized that preformation of a protein corona under controlled conditions ameliorates the protein corona effects, including colloidal stability in serum solutions. We report on the investigation of the stabilizing effects of a denatured bovine serum albumin (dBSA) protein corona formed on the surface of upconversion nanoparticles (UCNPs). UCNPs were chosen as a nanoparticle model due to their unique photoluminescent properties suitable for background-free biological imaging and sensing. UCNP surface was modified with nitrosonium tetrafluoroborate (NOBF_4_) to render it hydrophilic. UCNP-NOBF_4_ nanoparticles were incubated in dBSA solution to form a dBSA corona followed up by lyophilization. As produced dBSA-UCNP-NOBF_4_ demonstrated high photoluminescence brightness, sustained colloidal stability after long-term storage and the reduced level of serum protein surface adsorption. These results show promise of dBSA-based nanoparticle pretreatment to improve the amiability to biological environments towards theranostic applications.

## 1. Introduction

In the natural fluidic environment of a biological system, nanoparticles (NPs) experience chaotic adhesion of plasma proteins forming “protein corona” (PC) whose composition and kinetics determine the fate of NPs manifested as aggregation, clusterization, sedimentation due to gravity, and coupling to a flow [1]. The NP properties determine the protein binding profile that can cause the change in nanoparticle size, shape, surface charge. This biological coating determines NP interaction with biological milieu, including cellular uptake, safety profile [2] and residence time in blood circulation [3]. The residence time in blood circulation depends on the protein corona formed on NPs is recognized by the mononuclear phagocyte system–this process is termed “opsonization” and results in nanoparticle sequestration after the intravenous injection [4,5]. To endow NPs with stealth properties, surface engineering can reduce the adsorption of opsonins, evading the phagocyte system in vivo and aids development nanomedicine with less side effects caused by non-specific protein binding. Engineering NPs that can specifically bind certain proteins of interest (precoating the NPs with proteins or capturing specific serum proteins) for targeting purposes, can enhance the development of nanoparticles as drug delivery.

Upconversion nanoparticles (UCNPs) were chosen as a nanoparticle model due to their unique photoluminescent properties. Upconversion nanoparticles hold promise for broad applications in the life sciences due to their unique properties—conversion of near-infrared light into photoluminescence of the shorter wavelength (ultraviolet, visible or near-infrared) [6]. This property found a range of applications in highly sensitive biodetection by using near-infrared excitation to minimize the background signal and enhance the overall signal-to-noise ratio. Narrow multicolor emission, exceptional photostability, and low cytotoxicity represent the properties that confer UCNP advantageous in multiplexing and long-term detection in biological settings. UCNP were found nontoxic according to a broad range of experiments in vitro and in vivo, although several studies have shown variation in their cytotoxicity depending on their surface modifications [7,8,9]. The modest UCNP’s conversion efficiency has been improved by introducing homogenous core/shell crystal structure and tunable crystallinity [10,11]. These properties alongside exceptionally long photoluminescence (PL) lifetime (up to 1 ms) enabled suppression of the autofluorescence and backscatter signal interference by employing time-gated detection. This enabled to achieve biological tissue imaging penetration depth up to a centimeter [12,13,14].

The UCNP synthesis procedure has been improved to obtain well-formed, monodisperse and crystalline nanoparticles with high photoluminescence intensity. However, the outcome are lipophilic nanoparticles that need further surface modifications to become hydrophilic, biocompatible nanoparticles [15]. For example, capping UCNP with oleic acid (OA) plays a vital role in increasing the intensity of upconversion PL, preventing agglomeration, increasing colloidal stability, and controlling the size and morphology of nanoparticles [16]. These UCNP-OA are hydrophobic and require surface modification. One of the most successful examples is the ligand exchange method where OA is substituted with the inorganic compound nitrosonium tetrafluoroborate (NOBF_4_) with a molecular weight of 116.81 g/mol resulting in hydrophilic, relatively small and positively charged nanoparticles [17].

The enhanced uptake of nanoparticles into solid tumors is generally attributed to passive delivery, a route facilitated by the enhanced permeability and retention (EPR) effect [18]. The EPR effect is defined by the pathophysiology of tumor tissue, characterized by angiogenesis, hypervasculature, a defective vascular architecture and an impaired lymphatic drainage [19]. The premise is that nanoscale materials (smaller than 2000 nm) accumulate in solid tumor tissues much more than normal tissues as a result of their passive diffusion through the large pores of abnormal tumor vasculature [20]. Recently conducted series of experiments has challenged the current understanding of nanoparticle accumulation inside solid tumors. They have demonstrated that the inter-cellular gaps of the blood vessels are responsible for only 3–25% of nanoparticle extravasation through tumor vasculature and the actual dominant mechanism of nanoparticle transport is an active pathway through the trans-endothelial channels, inside fenestrae, or other mechanisms that are yet to be discovered [21].

The mere existence of the aforementioned protein corona phenomenon and its influence on NPs’ behavior and fate in biological environments have been realized quite recently [22,23]. PC-nanoparticle interaction is governed by hydrogen bonds, ionic and van der Waals forces, and hydrophobic interactions depending on surface properties of different nanomaterials [24]. In the last decade, realizing the inevitable presence of protein corona has presented an opportunity for exploiting this phenomenon by using proteins as stabilizers. Albumin, especially bovine serum albumin (BSA), has become the most intensively investigated protein as a stabilizer because as a natural component of the blood it is biocompatible, biodegradable, non-toxic and non-immunogenic [25,26]. Moreover, it is abundant and has a long circulation time in the blood [27].

In this work, we have assembled a photoluminescent nanocomplex based on UCNP modified with NOBF_4_ and coated with denatured BSA (dBSA) capable of up-converting near infrared (NIR) light into higher energy NIR and visible light. According to several studies dBSA promotes more stability and better inhibition of protein adsorption than native BSA [28,29,30]. Denaturing BSA at temperature 70 °C and above results in irreversible conformational changes of BSA’s tertiary and secondary structures turning them into primary structures [28]. Additionally, thermal denaturing increases the flexibility of the protein main chain as well as the stickiness and ability to adhere to hydrophilic surfaces [29,30]. On the other hand, BSA carries a negative net charge at pH above its isoelectric point (pH 4.7), the net charge of BSA is about –16 at pH 7.4 and the unfolding of BSA upon thermal denaturation renders it more negative, which helps strengthen the electrostatic interaction between dBSA and positively charged UCNP-NOBF_4_ [30,31,32]. The solubility and stability of the nanocomplexes in a colloidal suspension were confirmed and monitored three times over one year. The ability of dBSA coating to inhibit protein corona formation on the surface of dBSA-UCNP-NOBF_4_ in biological fluids was also evaluated.

## 2. Materials and Methods

### 2.1. Synthesis of UCNP-OA and UCNP-NOBF_4_

Hydrophobic UCNP-OA core/shell structure (NaY_0.794_F_4_:Yb_0.2_,Tm_0.06_/NaYF_4_) were synthesized by solvothermal decomposition method [33] in Federal Research Center “Crystallography and Photonics” of the Russian Academy of Sciences (RAS), Russia following the instruction of an earlier report [34]. The ligand exchange reaction was carried out to remove OA with NOBF_4_ (Sigma-Aldrich, St. Louis, MO, USA) rendering the nanoparticles hydrophilic according to the detailed procedure in an earlier report [35]. The obtained UCNP-NOBF_4_ were dispersed in deionized water and stored at 4 °C.

### 2.2. Characterization of UCNP-OA and UCNP-NOBF_4_

UCNP-OA were visualized using a transmission electron microscope LEO-912 ABOMEGA (Carl Zeiss, Oberkochen, Germany) for size and shape verification. The photoluminescence properties of the UCNP-OA and UCNP-NOBF_4_ were investigated using an SM2203 spectrofluorometer (SOLAR, Minsk, Belarus) and an ATC-C4000-200AMF-980-5-F200 external semiconductor laser module with a wavelength of 980 nm (Semiconductor devices, Saint-Petersburg, Russia). The PL emission spectra were recorded over the wavelength range 400–850 nm in a quartz cuvette with a 1 cm optical path length. The hydrodynamic diameters of UCNP-OA and UCNP-NOBF_4_ were acquired by dynamic light scattering (DLS), and the ζ-potential of UCNP-NOBF_4_ was determined by electrophoretic light scattering (ELS), using a Zetasizer Nano ZS system at 25 °C (Malvern Instruments Ltd., Malvern, UK) according to the manufacturer’s recommendations. FTIR absorption spectra were acquired using an IRPrestige-21 Fourier Transform Infrared Spectrophotometer (Shimadzu, Kyoto, Japan).

### 2.3. Optimization of dBSA Concentration for PC Formation on UCNP-NOBF_4_

BSA stock solution was prepared in deionized water by vigorous stirring using a magnetic stirrer and subsequent filtration through a membrane filter (0.22 μm) to eliminate protein clusters. The stock was then thermally denatured at 70 °C for 30 min. A dilution series of dBSA was prepared and incubated individually with UCNP-NOBF_4_ (1:1, v:v), where UCNP-NOBF_4_ was added dropwise to the dBSA suspension and vortexed. The ultimate concentrations in samples were 0.25 mg/mL of UCNP-NOBF_4_ and 5, 10, 25 and 50 μM of dBSA. The mixtures were incubated at room temperature for 4 h to allow maximum adsorption of dBSA onto the surface of UCNP-NOBF_4_ and the formation of a stable layer of dBSA protein corona [36]. The size and ζ-potential were measured 15 min and 4 h after incubation.

### 2.4. Lyophilization of dBSA-UCNP-NOBF_4_ and Evaluation of Their Colloidal Stability

Aqueous suspensions of dBSA-UCNP-NOBF_4_ complexes were obtained according to the optimum concentration determined in the previous section. Aliquots of dBSA-UCNP-NOBF_4_ complexes were washed twice with deionized water via sonication for 30 s and centrifugation at 10,000× *g* for 7 min, others were placed in glass vials, frozen at −80 °C and lyophilized using a FreeZone 6L freeze dryer (Labconco, Kansas, MO, USA). The vials were closed with rubber stoppers and sealed with Teflon. The samples were placed in a lyophilization chamber and kept at −50 °C for 24 h until the water was removed entirely. Lyophilized samples were stored at 4 °C. Later on, centrifuged samples were re-dispersed in deionized water via vortexing and sonication prior to DLS and ELS measurements, whereas lyophilized samples were re-dispersed in deionized water, phosphate-buffered saline pH 7.4 (PBS) buffer, Dulbecco’s Modified Eagle Medium (DMEM) and McCoy’s 5A cell culture media (HyClone Laboratories Inc., Logan, NE, USA), both supplemented with 10% fetal bovine serum (FBS) (HyClone Laboratories Inc., Logan, NE, USA) by vortexing and ultrasonication for 5 min. The hydrodynamic diameter of the dBSA-UCNP-NOBF_4_ complexes was measured by DLS in all the previously mentioned buffers at three checkpoints: one week, six months and one year after lyophilization.

### 2.5. Hard PC Formation on the Surface of UCNP-NOBF_4_ and Lyophilized dBSA-UCNP-NOBF_4_

Both UCNP-NOBF_4_ and lyophilized dBSA-UCNP-NOBF_4_ were incubated with complete cell culture medium (DMEM and McCoy’s 5A, each supplemented with 10% FBS) to form hard PC (final concentration of UCNP in each sample was 1 mg/mL) for 4 h under shaking at room temperature. The resulting colloids were centrifuged (10,000× *g*, 7 min) and re-dispersed with PBS by vortexing and sonication for 30 s. The colloids were washed three times with PBS and centrifuged to remove loosely bound proteins.

### 2.6. Protein Corona Quantification

The ability of dBSA coating on UCNP to block protein adsorption on to their surfaces in biological environments was analyzed and quantified by Pierce Micro BCA™ Protein Assay Kit (Thermo Fisher Scientific, Rockford, IL, USA) according to the manufacturer’s protocol. One sample of lyophilized dBSA-UCNP-NOBF_4_ (UCNP 1 mg/mL) was treated the same as the samples in Section 2.5 to characterize the dBSA background level and compare it with the protein corona formed in complete cell culture media. Later, 25 μL of each sample (dBSA-UCNP-NOBF_4_, UCNP-NOBF_4_@PC, and dBSA-UCNP-NOBF_4_@PC) was pipetted into five replicate wells of a 96-well plate. Next, 200 μL of micro-BCA working reagent was added per well. The plate was shaken on a rocking platform for 5 min at room temperature and then incubated for 60 min at 60 °C in dark conditions. The plate was cooled to room temperature prior to measuring the absorbance of the samples at 562 nm using a Synergy MX plate spectrophotometer (BioTek, Winooski, VT, USA) with the background absorbance subtracted. A series of bovine serum albumin solution at a concentration range of 0–100 μg/mL was prepared and treated the same way as the samples. A calibration curve was plotted and used to calculate the total protein concentration after background subtraction (Appendix A). The data obtained was analyzed using the GraphPad Prism 9.0 software (GraphPad Software, San Diego, CA, USA).

## 3. Results

### 3.1. Synthesis and Characterization of UCNP-OA and UCNP-NOBF_4_

UCNP-OA core/shell structure NaY_0.794_F_4_:Yb_0.2_,Tm_0.06_/NaYF_4_ were synthesized using the solvothermal decomposition method with firmly controlled size and shape and afterward converted into UCNP-NOBF_4_ by ligand exchange as illustrated in (Figure 1) [17,33]. Lanthanide and fluorine precursors were heated in an oxygen-free environment and later stabilized in an oleic acid solution, consequently cubic nanosized UCNP crystals were formulated (α-phase). In order to develop the more stable hexagon-shaped UCNP crystals (β-phase) additional heat treatment was applied [34]. The transition from α-phase to β-phase and subsequent construction of the inert NaYF_4_ shell remarkably increase the PL brightness of UCNP, a rather useful improvement for their bioimaging applications [37].

UCNP-OA investigation by TEM shows acquired hexagon-shaped nanoparticles with an average size of 35 ± 2.8 nm (Figure 2a). The hydrodynamic diameter of UCNP-OA and PDI showed stability and average polydispersity in hexane (Figure 2d) and (Table 1). These hydrophobic UCNP-OA are unfit for biological applications, and therefore were modified by ligand exchange to achieve hydrophilicity. The OA residues were substituted with the small ionic compound NOBF_4_ and yielded hydrophilic nanoparticles with a stable size distribution and a moderate polydispersity in aqueous solution (Figure 2d) and (Table 1). PDI values in Table 1 were both below 0.3 indicating a high size uniformity and minimal aggregation levels [38]. UCNP-NOBF_4_ nanoparticles had a positive ζ-potential as they are surrounded by NO^+^ ions on the surface and BF_4_^−^ as counterions (Figure 1) [17]. Their ζ-potential of +53.9 ± 0.551 mV at neutral pH (7.0) (Figure 3c) indicates high colloidal stability in aqueous suspension since the electrostatic repulsion between positively charged particles surpasses the Van der Waals interaction, lowering the possibility of aggregate formation [38]. The PL emission peaks of UCNP-OA and UCNP-NOBF_4_ are in the blue (478 nm) and NIR (800 nm) regions, corresponding to ^1^G_4_→^3^H_6_ and ^3^H_4_→^3^H_6_ transitions of Tm^3+^ ions, respectively, under excitation with 980 nm radiation (Figure 2b,c) [39]. The strong emission peak of these nanoparticles at 800 nm in the NIR region falls, within the transparency window of biological tissue (700–1100 nm), makes them ideal for bioimaging [6]. The PL intensity of UCNP-NOBF_4_ is significantly lower than UCNP-OA, nonetheless, the exhibited intensity is strong enough for bioimaging purposes [15]. FTIR absorption spectra of UCNP-OA and UCNP-NOBF_4_ were also acquired. The appearance of the 2933 and 2850 cm^−1^ peaks corresponds to the asymmetric and symmetric stretching vibrations of –CH_2_ groups of oleic acid, respectively (Appendix A). Additionally, two bands at 1560 and 1466 cm^−1^ are observed and assigned to the asymmetric and symmetric stretch of the COO^−^ of oleic acid, respectively. Moreover, the NOBF_4_ treatment of UCNP caused an intensity reduction of the peaks at 2933 and 2850 cm^−1^ and the appearance of a new band at 1085 cm^−1^ associated with BF_4_^−^ anions (Appendix A). These observations suggest the replacement of OA molecules by NOBF_4_.

### 3.2. Optimization of dBSA Concentration for PC Formation on UCNP-NOBF_4_

Protein corona of denatured BSA is formulated to stabilize UCNP-NOBF_4_ before further investigation. The objective is to form a stable layer of dBSA around UCNP-NOBF_4_ to shield the nanoparticles from aggregation and additional protein adsorption in biological fluids. Thermal denaturing of BSA at 70 °C induces irreversible unfolding of the BSA tertiary structure, increasing protein size and reducing the ζ-potential value (Appendix A) [30]. The concentration of dBSA is optimized to yield an albumin PC layer formed around UCNP-NOBF_4_ precisely and distributed evenly amongst the colloids in a manner that neither aggregations nor excess proteins are observed.

High dBSA concentrations maintained an excess of unbound proteins, whereas low dBSA concentrations led to relatively high level of aggregation (Figure 3a, Appendix A). The absolute value of ζ-potential increases with the rising dBSA concentration (Figure 3c). Size, ζ-potential, and colloidal stability of dBSA-UCNP-NOBF_4_ in suspension under all concentrations were sustained at room temperature for 4 h (Figure 3b,c, Appendix A) and (Appendix A) showing stability, consistency and integrity of the nanocomplex overtime. The concentration 10 µM of dBSA exhibited the best possible record of a single peak of nanoparticle’s hydrodynamic diameter distribution with a moderate polydispersity in a relatively stable state in colloidal suspensions (Table 2) and (Figure 3).

### 3.3. Centrifugation of dBSA-UCNP-NOBF_4_ and Evaluation of Their Colloidal Stability

Centrifugation as a precipitation approach for these nanocomplexes was proven ineffective as it promoted a high level of aggregation and sedimentation in deionized water. The hydrodynamic diameter was ~750 nm (Table 3) and (Figure 4). The majority of dBSA was removed by centrifugation and the retained portion of dBSA caused the ζ-potential value to deviate from naked UCNP-NOBF_4_ (Table 3) and (Figure 3) and rendered the colloids prone to agglomeration [38].

### 3.4. Lyophilization of dBSA-UCNP-NOBF_4_ and Evaluation of their Colloidal Stability

Another approach to achieve colloidal precipitation was conducted using lyophilization which ensured integrity preservation of the obtained dBSA-UCNP-NOBF_4_. These nanocomplexes were characterized in comparison with UCNP-NOBF_4_ and pure dBSA. PL emission of UCNP-NOBF_4_ was not repressed by the dBSA coating, on the contrary a slight increase of PL signal was observed after dBSA-coating because this additional layer of dBSA can reduce the quenching effect from water molecules resulting to a higher PL signal (Figure 5a) [40,41]. The hydrodynamic diameter confirms PC formation around the nanoparticles as the diagram shifted towards a larger size (Figure 5b) and the small peak corresponding to dBSA at about 10 nm is the product of lyophilization and ultrasonication. The FTIR spectrum of lyophilized dBSA-UCNP-NOBF_4_ shows a reduction in the 1085 cm^−1^ peak corresponding to the BF_4_^−^ anions and an alteration of the band at 1397 cm^−1^ assigned to amide III of dBSA, C≡N stretching mode and N–H bending mode (Appendix A). These remarks suggest a shielding of the BF_4_^−^ signal and interaction between the albumin and nanoparticles, leading to a successful dBSA coating.

Colloidal stability was assessed by re-dispersion of lyophilized dBSA-UCNP-NOBF_4_ in several buffers. The nanocomplex showed high stability in deionized water, PBS, and cell culture media supplemented with 10% FBS based on three replicate measurements by DLS (Figure 6). The small peaks (gray or dotted lines below 100 nm) in (Figure 6a,b) correspond to a small level of dBSA detachment and aggregation as a result of lyophilization and ultrasonication. Additionally, the multiple peaks (gray or dotted lines below 100 nm) in (Figure 6c,d) are explained by the corresponding signal of serum proteins and protein aggregates with a hydrodynamic diameter of about 10–60 nm in the presence of FBS. Any peaks above 300 nm are related to a minor level of aggregation (Figure 6a,d) below 0.5%. Therefore, with regard to the previous section, lyophilization was revealed to be more effective than centrifugation in maintaining the stability of UCNP-NOBF_4_ coated with denatured BSA in the presence of serum proteins, which is consistent with the literature and albumin’s ability to act as a lyoprotectant [42]. These nanocomplexes were stored at 4 °C in glass vials for further assessment of their integrity and stability.

### 3.5. Colloidal Stability after Long-Term Storage

The nanocomplexes were evaluated again after six months of their lyophilization, only this time they were re-dispersed by vortexing without ultrasonication. The outcome displayed colloidal stability and consistency with the previous observation in Section 3.4. Although, it is clear now that lyophilization itself was responsible for a certain level of dBSA detachment from the nanoparticles (Figure 7a,b). Additionally, ultrasonication doesn’t affect the colloidal stability of the nanocomplex nor their integrity, but without it a higher level of aggregation (above 300 nm) seems to appear in different buffers (Figure 7b–d). 

Finally, after a year has passed with the lyophilized nanocomplexes stored at 4 °C, their colloidal stability was reassessed in the same buffers as above, and the DLS measurements were acquired before (Figure 8) and after ultrasonication (Figure 9). In both cases, the observations were similar to the six months stability evaluation (Figure 7) and consistent with Section 3.4. Nonetheless, the sustained stability and retained quality of the nanocomplexes shows actual potential for further investigation and application. It is clear that ultrasonication relieves the level of aggregation observed without it even if it is not essential for the actual re-suspension procedure of the nanocomplexes.

The average results of all four data sets of stability evaluation show collectively that the dBSA-UCNP-NOBF_4_ corresponding peaks (100–300 nm) overlap in all different buffers (Figure 10). The data analysis at all three checkpoints express the same conclusion, lyophilization and further storage at 4 °C for a year have no influence on the level of dBSA detachment, the integrity or the colloidal stability of the obtained complexes.

### 3.6. Hard Protein Corona Quantification

The influence of dBSA coating of UCNP-NOBF_4_ on protein adsorption in biological fluids was evaluated by micro-BCA assay. The predetermined concentration of dBSA in each lyophilized sample is 10 μM, and the sustained portion of dBSA after three cycles of washing with PBS was about 7 μg/mL. The micro-BCA analysis has shown a significant decrease in protein adsorption on the surface of UCNP coated with dBSA with comparison to naked UCNP. Additionally, these observations were repeated upon incubation in two different cell culture media (Figure 11).

## 4. Discussion

Recent advances in nanobiomedical research are emerging rapidly and breaking boundaries that justified approved applications of many nanoparticles in preclinical and clinical trials [43]. UCNPs are amongst the most promising candidates for creating multifunctional nanocomposites for improved bioimaging, theranostics, targeted and combined therapies. Approaches to stabilize nanoparticles using albumin to increase circulation time and block the formation of unwanted protein corona have been investigated on several nanoparticle types [27,44]. Albumin coating generally provides better water-solubility, increased bioavailability, enhanced biocompatibility, reduced toxicity, reduced protein corona formation, enhanced cellular uptake and improved blood circulation time, compared with naked nanoparticles [29,45]. The passive pathway of tumor-targeting has been the critical factor in nanoparticle delivery for decades as it can be applied to large tumors by systemic intravenous injection. Recent discoveries have shed the light on more effective active transport pathways in tumor blood vessels that account for the majority of nanoparticle accumulation inside the tumor microenvironment [21]. Whether it is via the EPR-effect or via active trans-endothelial transport, the fact remains that the longer nanoparticles circulation in the blood, the higher propensity for them to reach the tumor site and accumulate in it [20,21]. Generally speaking, albumin coating takes advantage of both because it allows longer circulation in the blood stream and endocytosis facilitated by albumin receptors such as gp60 [19,27].

The present study demonstrated the assembly of a new photoluminescent UCNP-based nanocomplex coated with NOBF_4_ and stabilized with a layer of dBSA protein corona. Determination of dBSA optimal concentration was essential to ensure the best ratio between dBSA and UCNP-NOBF_4_ for nanoparticle stabilization and protein corona formation. The dBSA layer of proteins is meant to be as fixed as possible to evade any undesirable colloidal aggregation and minimize the amount of unbound dBSA molecules. A mixture of UCNP-NOBF_4_ and dBSA stimulates dBSA protein corona formation around UCNP-NOBF_4_ immediately within 15 min and sustains it over a period of 4 h until lyophilization (Figure 3, Appendix A). The latter process is protein concentration dependent, an important and recently established factor in protein corona formation, and it is supported by the displayed results in (Figure 3, Appendix A) [46]. The displayed results also demonstrate centrifugation inadequacy to isolate dBSA-UCNP-NOBF_4_ nanocomplexes as well as lyophilization efficiency for the precipitation and stabilization of the same nanoparticles coated with denatured BSA (Figure 5) and (Figure 6). The current approach benefits from the strong electrostatic interactions between positively charged UCNP-NOBF_4_ and negatively charged dBSA under physiological pH (Figure 3c) as well as the excellent lyoprotectant capacity of albumin itself [42]. Furthermore, the obtained nanocomplexes have displayed excellent colloidal stability after a year of storage at 4 °C, which is a remarkable feature for any potential biological or medical application.

The dBSA coating is strongly attached to the nanoparticles as it tolerated three cycles of washing with PBS and repetitive centrifugation. This strong attachment has led to a diminished amount of serum protein adsorption from cell culture media onto the UCNP surfaces with reference to the control naked UCNP. This ability enhances nanoparticle effectiveness as it reduces the undesired effects of protein corona formation upon entering biological environments.

Colloidal stability of these nanocomplexes in various buffers has been demonstrated in the presence and absence of serum protein up to one year of their lyophilization. These nanocomplexes reduce the adsorption of serum proteins onto nanoparticles and had a high PL intensity in the NIR region within the transparency window of living tissues. Therefore, they make an excellent platform to create a multifunctional nanocomposite for further investigation and more effective medical applications.

## Figures and Tables

**Figure 1 materials-14-01657-f001:**
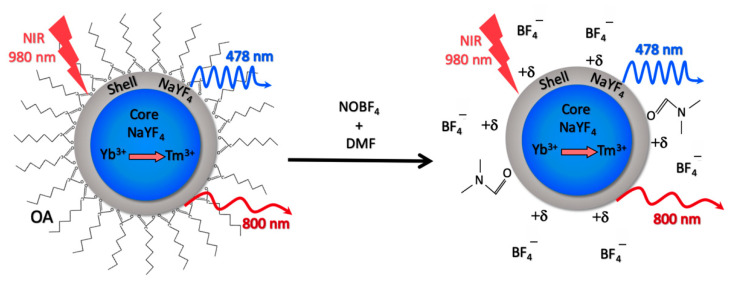
A schematic illustration of upconversion nanoparticles oleic acid (UCNP-OA) core/shell structure and the ligand exchange method converting UCNP-OA into UCNP-NOBF_4_. DMF—dimethylformamide.

**Figure 2 materials-14-01657-f002:**
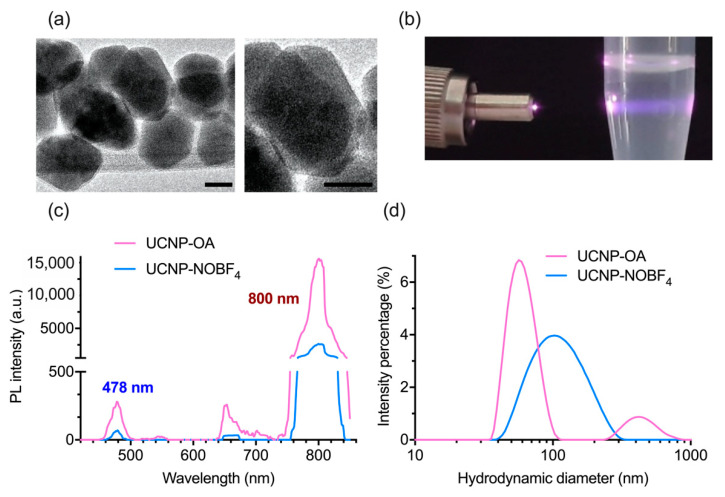
Characterization of UCNP-OA core/shell structure (NaY_0.794_F_4_: Yb_0.2_, Tm_0.06_/NaYF_4_) and UCNP-NOBF_4_: (**a**) TEM image of UCNP-OA, scale bar 20 nm; (**b**) UCNP in suspension illuminated by a laser with a wavelength of 980 nm; (**c**) photoluminescence (PL) emission spectrum of UCNP-OA and UCNP-NOBF_4_ under excitation at 980 nm; (**d**) hydrodynamic diameter distributions of UCNP-OA and UCNP-NOBF_4_ acquired by DLS in hexane and deionized water, respectively.

**Figure 3 materials-14-01657-f003:**
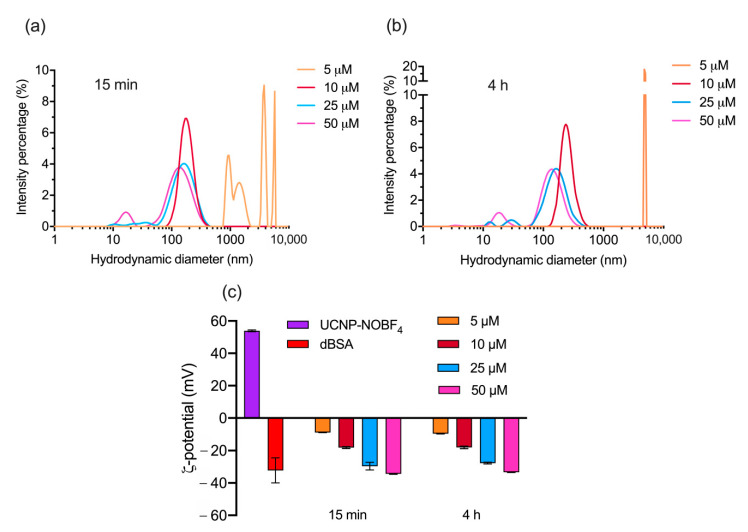
Concentration optimization of dBSA for forming PC on the surface of UCNP-NOBF_4_: (**a**,**b**) hydrodynamic diameter of dBSA-UCNP-NOBF_4_ following the incubation of UCNP-NOBF_4_ (0.25 mg/mL) with dBSA at different concentrations (5–50 μM) at room temperature for 15 min and 4 h, respectively measured by DLS; (**c**) the ζ-potential of UCNP-NOBF_4_, dBSA, and dBSA-UCNP-NOBF_4_ recorded by electrophoretic light scattering (ELS). All samples were dispersed in deionized water.

**Figure 4 materials-14-01657-f004:**
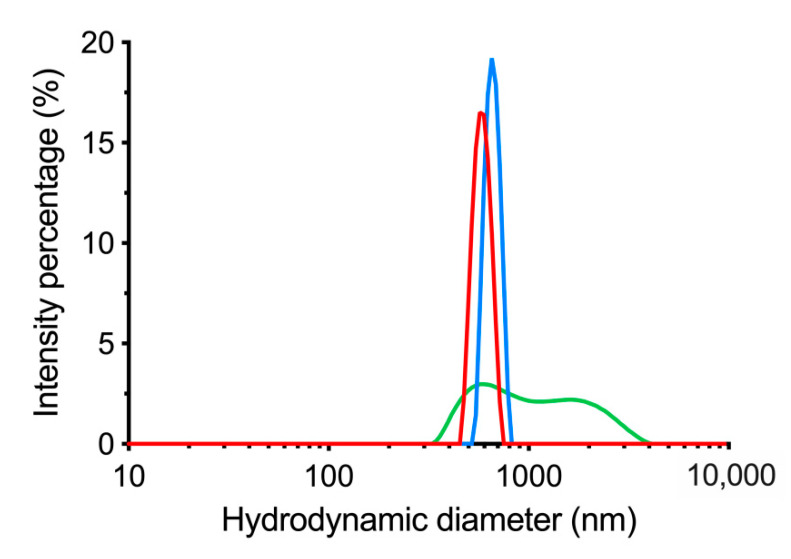
Hydrodynamic diameter of dBSA-UCNP-NOBF_4_ after using centrifugation. dBSA-UCNP-NOBF_4_ were measured by DLS following the incubation of UCNP-NOBF_4_ (0.25 mg/mL) with dBSA (concentration over 50 μM) and centrifuged at 10,000× *g* for 7 min. DLS three repeated size measurements are shown in blue, red, and green.

**Figure 5 materials-14-01657-f005:**
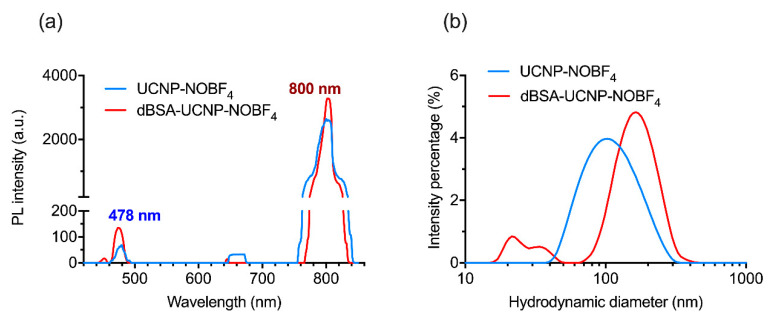
Characterization of UCNP-NOBF_4_ and lyophilized dBSA-UCNP-NOBF_4_ (dBSA at optimal concentration—10 μM): (**a**) PL emission spectrum of UCNP-NOBF_4_ and lyophilized dBSA-UCNP-NOBF_4_ under excitation at 980 nm; (**b**) hydrodynamic diameter distributions of UCNP-NOBF_4_ and lyophilized dBSA-UCNP-NOBF_4_ acquired by DLS in deionized water.

**Figure 6 materials-14-01657-f006:**
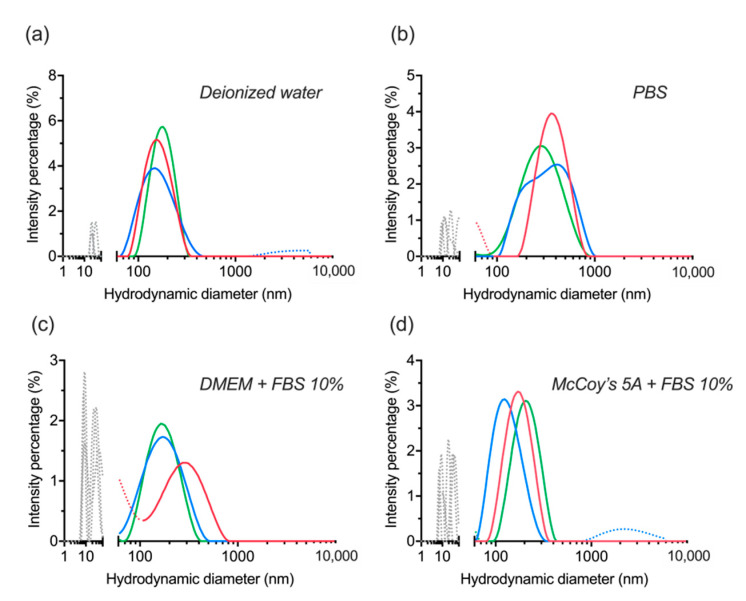
Evaluation of colloidal stability of lyophilized dBSA-UCNP-NOBF_4_ in different buffers after one week of lyophilization and mixed by vortexing and ultrasonication, based on the results of three repeated size measurements (shown in blue, red, and green) by DLS: (**a**) in deionized water; (**b**) in PBS; (**c**) in DMEM cell culture medium supplemented with 10% FBS; (**d**) in McCoy’s 5A cell culture medium supplemented with 10% FBS. Gray and/or dotted lines below 100 nm correspond to the presence of proteins and protein aggregates; dotted lines over 300 nm are aggregates.

**Figure 7 materials-14-01657-f007:**
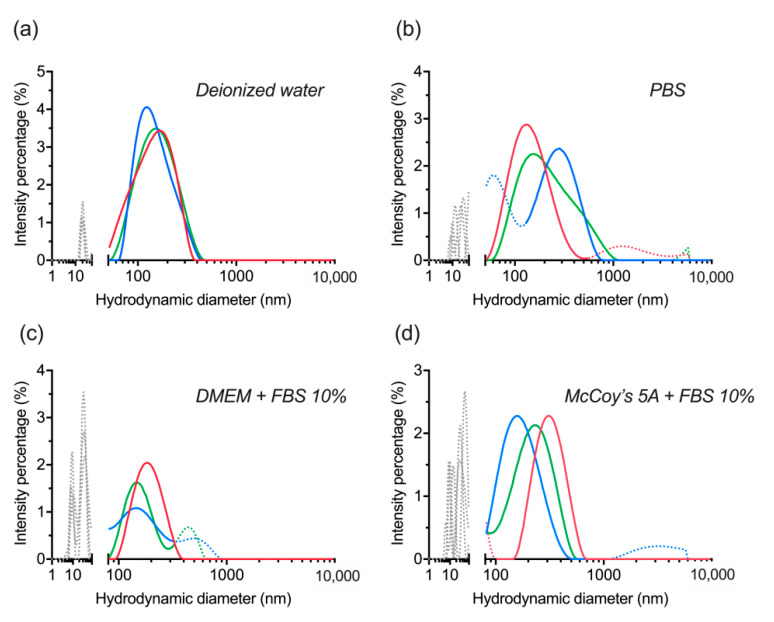
Evaluation of colloidal stability of lyophilized dBSA-UCNP-NOBF_4_ in different buffers after six months of lyophilization and mixed by vortexing only, based on the results of three repeated size measurements (shown in blue, red, and green) by DLS: (**a**) in deionized water; (**b**) in PBS; (**c**) in DMEM cell culture medium supplemented with 10% FBS; (**d**) in McCoy’s 5A cell culture medium supplemented with 10% FBS. Gray and/or dotted lines below 100 nm correspond to the presence of proteins and protein aggregates; dotted lines over 300 nm are aggregates.

**Figure 8 materials-14-01657-f008:**
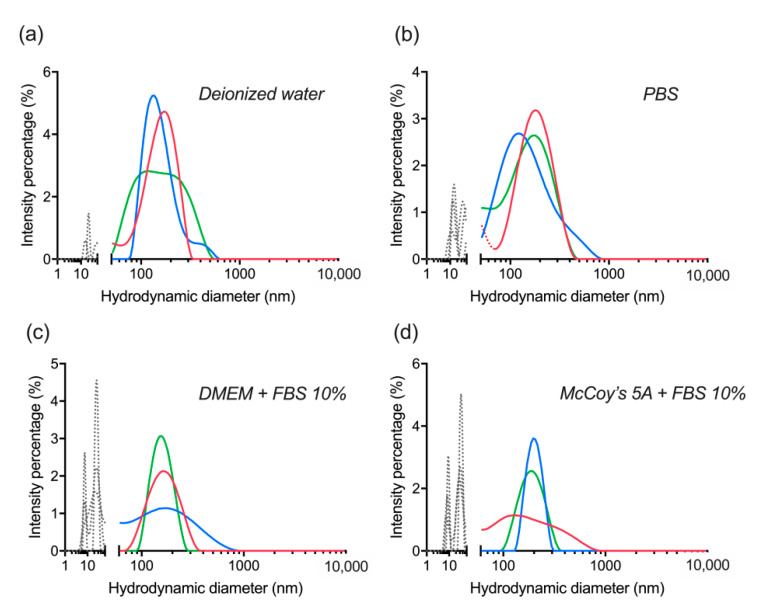
Evaluation of colloidal stability of lyophilized dBSA-UCNP-NOBF_4_ in different buffers after one year of lyophilization and mixed by vortexing only, based on the results of three repeated size measurements (shown in blue, red, and green) by DLS: (**a**) in deionized water; (**b**) in PBS; (**c**) in DMEM cell culture medium supplemented with 10% FBS; (**d**) in McCoy’s 5A cell culture medium supplemented with 10% fetal bovine serum (FBS). Gray and/or dotted lines below 100 nm correspond to the presence of proteins and protein aggregates; dotted lines over 300 nm are aggregates.

**Figure 9 materials-14-01657-f009:**
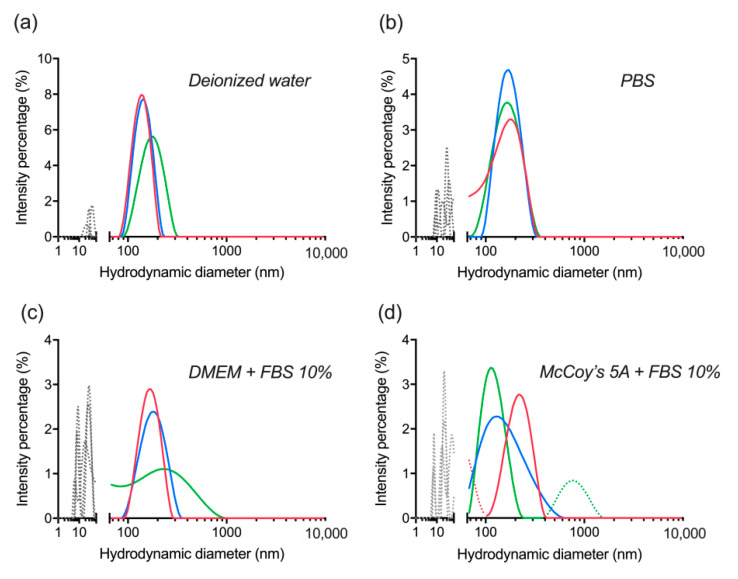
Evaluation of colloidal stability of lyophilized dBSA-UCNP-NOBF_4_ in different buffers after one year of lyophilization and mixed by vortexing and ultrasonication, based on the results of three repeated size measurements (shown in blue, red, and green) by DLS: (**a**) in deionized water; (**b**) in PBS; (**c**) in DMEM cell culture medium supplemented with 10% FBS; (**d**) in McCoy’s 5A cell culture medium supplemented with 10% FBS. Gray and/or dotted lines below 100 nm correspond to the presence of proteins and protein aggregates; dotted lines over 300 nm are aggregates.

**Figure 10 materials-14-01657-f010:**
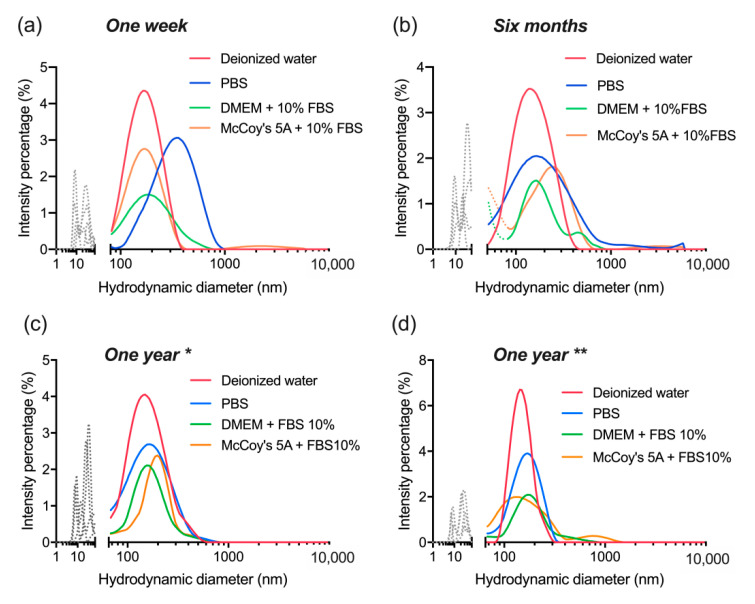
Evaluation of colloidal stability of lyophilized dBSA-UCNP-NOBF_4_ in different buffers at all three checkpoints based on the average results of three repeated size measurements by DLS: (**a**) after one week of lyophilization; (**b**) after six months; (**c**,**d**) after one year. * before ultrasonication, ** after ultrasonication; gray and/or dotted lines below 100 nm correspond to the presence of proteins and protein aggregates.

**Figure 11 materials-14-01657-f011:**
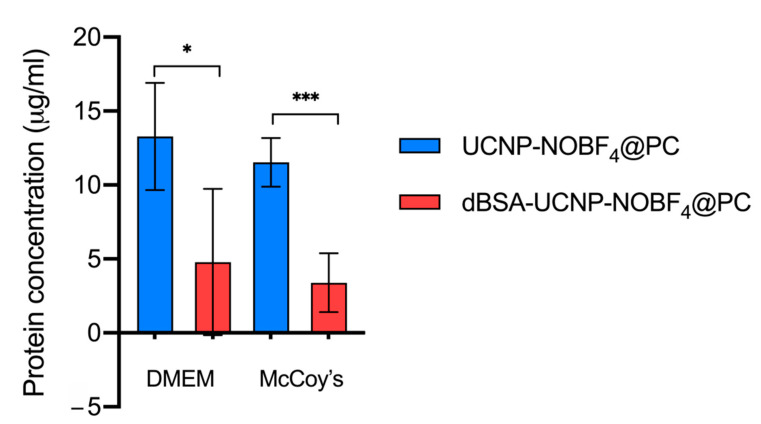
Hard protein corona quantification by micro-BCA assay. All samples were treated identically and subject to three cycles of washing with PBS. Protein corona formation on the surface of lyophilized dBSA-UCNP-NOBF_4_ was calculated after eliminating the dBSA background. The results were analyzed with an unpaired t-test (n = 5; *: *p* < 0.05, ***: *p* < 0.001) using GraphPad Prism 9.

**Table 1 materials-14-01657-t001:** The hydrodynamic diameter of UCNP-OA and UCNP-NOBF_4_ acquired by dynamic light scattering (DLS) in hexane and deionized water, respectively.

UCNP	Average Size, nm	PDI
UCNP-OA	67.94 ± 0.38	0.254 ± 0.011
UCNP-NOBF_4_	101 ± 0.473	0.169 ± 0.011

**Table 2 materials-14-01657-t002:** The hydrodynamic diameter of dBSA-UCNP-NOBF_4_ nanocomplexes at different dBSA concentrations (5–50 μM) measured by DLS in deionized water.

UCNP-NOBF_4_ (0.25 mg/mL) Incubated with dBSA	Average Size, nm	PDI
Incubation time: 15 min		
5 μM	4755 ± 786.6	0.281 ± 0.159
10 μM	176.2 ± 2.937	0.127 ± 0.006
25 μM	112.3 ± 4.994	0.375 ± 0.071
50 μM	80.27 ± 0.6862	0.56 ± 0.008
Incubation time: 4 h		
5 μM	7721 ± 383.9	0.13 ± 0.024
10 μM	247 ± 5.918	0.139 ± 0.016
25 μM	116.3 ± 2.468	0.333 ± 0.003
50 μM	78.82 ± 0.8603	0.569 ± 0.005

**Table 3 materials-14-01657-t003:** The hydrodynamic diameter and ζ-potential of dBSA-UCNP-NOBF_4_ after centrifugation acquired by DLS and ELS, respectively.

Centrifuged dBSA-UCNP-NOBF_4_		
Average Size, nm	PDI	ζ-Potential
747.5 ± 76.71	0.292 ± 0.015	+6.74 ± 1.21

## Data Availability

The datasets generated and analyzed during the current study are available on request from the corresponding author.

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
