# Peer review of "Controlled Formation of a Protein Corona Composed of Denatured BSA on Upconversion Nanoparticles Improves Their Colloidal Stability"

_materials, 2021, doi:10.3390/ma14071657_

Round 1
Reviewer 1 Report
The manuscript deals with surface engineering of upconversion nanoparticles through the coating with denatured BSA and evaluates the effect on the short-term and long-term colloidal stability and the formation of the hard protein corona. The design of protein corona of nanoparticles for biomedical applications is a very important topic. Overall the manuscript is well written and well structured. The results are discussed adequately. There are only a few aspects that need to be improved before being considered for publication:
-Figure 5-There is an increase in PL intensity in lyophilized coated NPs. Was this increase also observed in the coated NPs before lyophilization? Can the authors explain this increase?
-Line 206: please indicate pH value (the zeta potential values vary with pH)
-line 105 : “about –16 ” units are missing
Reviewer 2 Report
This is a really interesting article presented by the authors as protein adsorption is a really big problem in the clinical translation of NPs.
That said i would argue that the title is a little bit misleading as the long term colloidal stability of these NPs has not been studied, more like the shelf life stability. More specifically the largest period studied was 4 hours, and i would argue that the results are a little alarming as there is an almost 50% change in the 10%dBSA NPs hydrodynamic diameter used for further studies. Maybe a 48h study would make it easier to draw the right conclusions.
Furthermore the 10% dBSA NPs seem to present a z-potential lower than -20mV and in addition with the presence of proteins and protein aggregates could lead to ostwald ripening processes and consequently to further aggregation.
It is my opinion that in this article the shelf life stability has been successfully studied and not the long term colloidal stability. In order to establish long term colloidal stability more measurements should be performed in the colloidal state in different time frames, e.g 2h - 4h - 8h - 24h, etc. Polydispersity index and Z-pot measurements should also be included in the long term colloidal stability study.
Thank you.
Reviewer 3 Report
The paper by Shanwar et al is interesting, clearly written and well performed.
I have a minor comment:
Did the authors try different temperatures or pH for their experiments? If not, can they elaborate on those variables as well?
Reviewer 4 Report
In this work, the authors have studied the effect of precoating with denatured BSA of upconverting nanoparticles on their opsonization in physiological ebvironments. The UCNP have been synthesized according to established procedures and rendered hydrophilic by ligand exchange with nitrosonium tetrafluoroborate (NOBF4). It is demonstrated that the dBSA precoating prevents to significant extent adsorption of proteins from the physiological medium to the UCNP. fThis result is of certain interest with view to eventual biomedical UCNP applications and it has been previously demonstrated by other groups for other types of nanoparticles.
However, the work suffers from drawbacks that might influence unfavorably the results:
- In contrast to previous studies utilizing denatured serum albumin for nanoparticle precoating, BSA is denatured prior to its addition to the UCNP. However, it is well known that proteins, and serum albumin in particular, display a tendency to aggregate upon their thermal denaturation. The authors do not mention did they observe BSA aggregation upon denaturation and how the denatured BSA aggregates might have influenced their results. This problem needs to be addressed before reaching any conclusions on the validity of the reported results.
- The preparation procedure is not described in sufficient detail and needs to be substantially complemented. For example, it remains unclear what is the amount of unbound dBSA in the different preparations and does it have any influence on the results.
Other comments:
- the authors should explain in more detail how native BSA compares with denatured BSA and why did they choose denatured BSA for their study.
- the number of figures with size measurements appears to be excessive.
-line 208 "gravitational attraction force": The attraction force between the NPs is certainly not of gravitational origin. That needs to be corrected.
- Table 1. Although they are in different media, it remains unclear why the UCNP-OA in hexane are much smaller than the UCNP-NOBF4 in deionized water. The authors should comment on this difference.
- abbreviations like DMF in Fig. 1 and PC need to be explained or avoided.
- language is generally good, although some improvement is certainly in order
Round 2
Reviewer 2 Report
All of the points made during the first round of review were adequately addressed by the authors.
Author Response
The authors thank the reviewer for their comments to improve the manuscript. We are glad that we were able to adequately respond to all comments.